# LI-DWT- and PD-FC-MSPCNN-Based Small-Target Localization Method for Floating Garbage on Water Surfaces

**Ping Ai** [1,2], **Long Ma** [2,*] and **Baijing Wu** [3]

1    College of Hydrology and Water Resources, Hohai University, Nanjing 210098, China; aip@hhu.edu.cn
2    College of Computer and Information Engineering, Hohai University, Nanjing 211100, China
3    School of Electronic and Information Engineering, Lanzhou Jiaotong University, Lanzhou 730070, China; 12211816@stu.lzjtu.edu.cn
*    Correspondence: malong@mail.lzjtu.cn; Tel.: +86-13919333385

**Abstract:** Typically, the process of visual tracking and position prediction of floating garbage on water surfaces is significantly affected by illumination, water waves, or complex backgrounds, consequently lowering the localization accuracy of small targets. Herein, we propose a small-target localization method based on the neurobiological phenomenon of lateral inhibition (LI), discrete wavelet transform (DWT), and a parameter-designed fire-controlled modified simplified pulse-coupled neural network (PD-FC-MSPCNN) to track water-floating garbage floating. First, a network simulating LI is fused with the DWT to derive a denoising preprocessing algorithm that effectively reduces the interference of image noise and enhances target edge features. Subsequently, a new PD-FC-MSPCNN network is developed to improve the image segmentation accuracy, and an adaptive fine-tuned dynamic threshold magnitude parameter $V$ and auxiliary parameter $P$ are newly designed, while eliminating the link strength parameter. Finally, a multiscale morphological filtering postprocessing algorithm is developed to connect the edge contour breakpoints of segmented targets, smooth the segmentation results, and improve the localization accuracy. An effective computer vision technology approach is adopted for the accurate localization and intelligent monitoring of water-floating garbage. The experimental results demonstrate that the proposed method outperforms other methods in terms of the overall comprehensive evaluation indexes, suggesting higher accuracy and reliability.

**Keywords:** image segmentation; target location; water-floating garbage; discrete wavelet transform; pulse-coupled neural network; human visual lateral inhibition network

## 1. Introduction

### 1.1. Background

With extensive population growth and rapid economic and social developments along rivers, lakes, and coastal areas worldwide, water-floating garbage has severely threatened anthropogenic and other natural ecosystems. Here, water-floating garbage refers to man-made solid waste or natural objects floating on rivers, lakes, and seas [1]; it differs from other types of floating objects in terms of its shape and floating movement characteristics. In this study, we consider small-target [2] water-floating garbage as our research object. We use computer vision technology to address poor localization accuracy under conditions of illumination, water waves, and complex backgrounds; moreover, we improve the ability of the visual tracking and position prediction of small-target water-floating garbage [3].

Thus far, research on water-floating garbage monitoring using computer-vision technology has primarily focused on intelligent real-time target localization and classification to accurately obtain the category and location information of water-floating garbage. Following this, the intelligent tracking, localization, pollution assessment, salvaging, and treatment of water-floating garbage are performed based on this information. With the

advent of big data, neural-network-based image vision and machine-learning techniques have been applied extensively to the research on intelligent positioning and classification of water-floating garbage [4–6]. These techniques can be classified into image-detection- and image-segmentation-based localization. Among these, image-detection-based localization is a computational processing method that uses image pixel blocks as the minimum processing units, and substantial recent research in the field has primarily focused on deep learning. For instance, Yi et al. [7] improved the faster regions with convolutional neural network (Faster R-CNN) method by incorporating a class activation (CA) network. The CA network could reduce localization errors without affecting the recognition accuracy, while effectively detecting and locating floating objects on water surfaces. Similarly, van Lieshout et al. [8] proposed a system capable of automatically monitoring plastic pollution through deep learning in five different rivers. Li et al. [9] proposed a garbage detection method based on the "You Only Look Once" (YOLO) version 3 backbone network to improve the accuracy of water-floating garbage detection. The authors employ a new approach of setting the corresponding anchor frame size on a homemade dataset and applying it to a water-floating robot. Armitage et al. [10] proposed a YOLOv5-based model for marine plastic garbage detection and distribution assessment to more effectively assess the abundance and distribution of global macroscopic oceanic floating plastic. The results reveal that their method could effectively detect plastic garbage with different sizes and obtain its location information.

Conversely, image-segmentation-based localization is a computational processing method that uses image pixel points as the smallest processing units. This localization method obtains more accurate target contour and position coordinate information, which is particularly advantageous for the high-precision localization problem of small-sized water-floating garbage. Notably, latest research on this method are focused on machine learning and neural networks. For example, Arshad [11] proposed an algorithm capable of effectively detecting and monitoring multiple ships in real time using morphological operations and edge information to segment and locate ships for ship detection and tracking, in addition to a smoothing filter and Sobel operator for edge detection. Imtiaz et al. [12] used the intensity and temporal probability maps of input image frames; these are then combined to determine the threshold for segmenting driftwood targets in water using the temporal connection method, thereby effectively overcoming the effects of illumination changes and wave interference for rapid driftwood target detection in videos. Similarly, Ribeiro et al. [13] used the YOLACT++ segmentation network with ResNet50 as the backbone feature extraction network and implemented a 3D constant rare factor (CRF) to improve frame loss and ensure the temporal stability of the model. Moreover, they constructed a synthetic floating ship dataset that aided the localization of ship targets; however, it increases the computational burden. Li et al. [14] proposed an improved Otsu method based on the uniformity measure to segment water surfaces, achieving adaptive selection of thresholds through the uniformity metric function to better segment water surfaces. Jin et al. [15] proposed an improved Gaussian mixture model (GMM) based on the automatic segmentation method (IGASM) to monitor water surface floaters. The IGASM improves the background update strategy; maps the GMM results based on the hue, saturation, and value of the color space; and applies the light and shadow discriminant function to solve the light and shadow problem. The extracted foregrounds are smoothed using morphological methods, and the IGASM method is verified on six video datasets. Water surface floaters in the videos are detected rapidly and accurately by mitigating the effects of light, shadow, and ripples on the water surface.

### 1.2. Related Research

The image segmentation-based location method has always been a focus of image vision and machine learning applications, and represents an important component of image processing analysis methods. Compared with other fields of image segmentation, the small-target images of floating garbage on water surfaces are characterized by complex

backgrounds and external environmental interference such as light and water waves [16] in the segmentation process. As opposed to large/medium-sized targets, the small-size target edge segmentation offset produced by one pixel point has a significant impact on the segmentation and localization accuracy; however, edge segmentation is more difficult in this case.

In this study, we use the discrete wavelet transform (DWT), which is known to be robust and to effectively suppress noise in both the time and frequency domains. We perform discrete wavelet decomposition and the reconstruction of images with complex backgrounds and interference from illumination and water waves, which could be effectively filtered out. In a previous study, Fujieda et al. [17] performed wavelet multilevel decomposition of the original image and spliced the low-frequency, high-frequency, and air domain features. Following fusion, these features are added to a CNN to compensate for the lost spectral information, in turn improving the texture classification accuracy. Li et al. [18] used the Daub5/3 algorithm to improve the multilevel decomposition processing of the wavelet transform on images and achieve multiscale image enhancement. Zhang et al. [19] combined the DWT with two-dimensional multilevel median filtering and proposed an adaptive remote-sensing image denoising algorithm that could adaptively select the threshold and improve the denoising ability of the wavelet transform. Fu et al. [20] combined the DWT with a generative adversarial network to construct a two-branch network for image enhancement. The proposed method prevented the loss of texture details, reducing convergence difficulty during training in haze image enhancement, and obtaining good evaluation results on the RESIDE dataset. To reduce the interference of changes in illumination on image recognition, Liang et al. [21] proposed a new framework based on the wavelet transform and principal components to improve the accuracy of face recognition under illumination changes. They used a particle swarm optimization neural network for face recognition and experimentally demonstrated robust visual effects under different illumination conditions, along with significantly improved recognition performance.

The pulse-coupled neural network (PCNN) model has been used extensively in image segmentation and other processing fields [22–25]. The nonlinear computation and neural ignition method of the PCNN model can refine target edge features during water-drifting garbage image segmentation and achieve more accurate segmentation. However, previous research on this method has primarily focused on medical applications while largely ignoring its applications in the field of water-drifting garbage image segmentation. For example, Guo et al. [26] improved the PCNN by integrating a spiking cortical model to achieve coarse-to-fine mammography image segmentation. Yang et al. [27] changed the popular simplified PCNN (SPCNN) model to an oscillating sine–cosine pulse-coupled neural network (SCHPCNN) and obtained good image quantization results. However, the complex mechanism and parameter settings of the PCNN are known to considerably limit the PCNN algorithm. In response, Deng et al. [28] analyzed the relationship between PCNN network parameters and mathematical coupling in fire extinguishing, coupling of adjacent neurons, and convergence speed of the PCNN. Consequently, they proved the basic law of neuron extinguishing in the PCNN, achieved the best comprehensive performance, and overcame the associated limitations. However, as the conventional PCNN model uses grayscale features of images as stimuli inputs, it cannot satisfy the requirements of image processing on the human vision system (HVS); thus, adoption of the HVS can result in more refined image processing [29]. Lian et al. [30] combined the characteristics of the PCNN and HVS and proposed an MSPCNN that could segment medical images of gallstones more accurately by changing the stimulus input. Similarly, Yang et al. [31] proposed a saliency-motivated improved simplified PCNN (SM-ISPCNN), which was validated on mammograms from the Gansu Cancer Hospital. The results indicate that the model demonstrated significant potential for clinical applications.

## 2. Materials and Methods

### 2.1. Dataset and Description

In the absence of publicly available datasets, we prepare a dataset for our experiment by recording data at the Yanbai Yellow River Bridge in Chengguan District, Lanzhou City (latitude 36.07484583, longitude 103.8843389). As a representative inland river, the Yellow River is characterized by a rapid flow, low visibility, and a complex riparian environment; it is also subject to water wave motion and illumination. Furthermore, owing to the complex and diverse riparian environment, a large drop between the river surface and river bank, and given the relatively wide river surface, the litter floating on its surface often appears as a small target in monitoring images. Therefore, the collection environment exhibits the typical characteristics of the problem that is investigated in this study and is suitable for this experiment.

As shown in Figure 1, seven data collection points are established. A model Canon PowerShot SX730 HS camera with a fixed angle is set up for the entire process of filming water-floating garbage at all points. The collected video data are extracted as jpg images according to the frames, filtered out into three different filming background datasets, and divided into datasets 1, 2, and 3 with 200 images each, where dataset 3 is a multitarget dataset of water-floating garbage. The image size is 5184 × 2912 pixels, with a horizontal as well as vertical resolution of 180 dpi, bit depth of 24, and a resolution unit of an inch. The trash target size accounted for a relatively small percentage relative to the image background size. The target percentage in the dataset images ranged from 0.13% to 0.60%, which belongs to the category of small targets [2], and is disturbed by riverbank reflections, lighting, and water waves to different degrees, in order to represent the floating garbage more clearly, the floating garbage is marked with a yellow box. As illustrated in Figure 2.

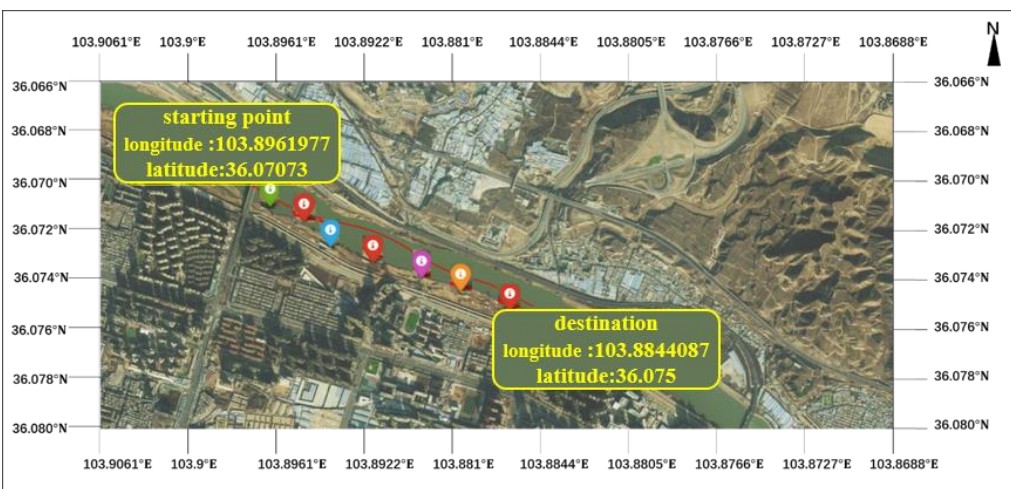

**Figure 1.** Yellow River data sampling location. Different colourful tags stands seven data collection points.

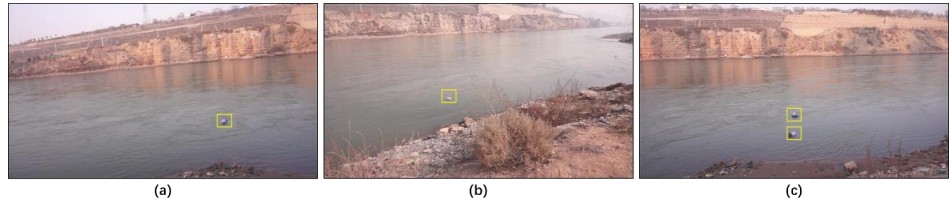

**Figure 2.** Example data plots: (**a**) dataset 1, (**b**) dataset 2, and (**c**) dataset 3. The yellow box is the marker box for floating garbage.

The latitude and longitude were acquired using the Galaxy 1-GNSS of the Southern Satellite Navigation Instrument Company, which adopts the RTK measurement system, with the following parameters: equipment positioning accuracy horizontal: 0.25 m + 1 ppm;

RMS vertical: 0.50 m + 1 ppm RMS SBAS; differential positioning accuracy: typical <5 m 3DRMS Static GNSS; measurement plane accuracy: ±2.5 mm + 1 ppm; elevation accuracy: ±5 mm + 1 ppm; real-time dynamic measurement plane accuracy: ±8 mm + 1 ppm; and elevation accuracy: ±15 mm + 1 ppm.

### 2.2. Methods

The proposed LI-DWT- and PD-FCMSPCNN-based localization method for small-target images of water-floating garbage comprises three steps: preprocessing, segmentation, and postprocessing, as shown in Figure 3. The algorithm consists of the following steps. See Appendix A.

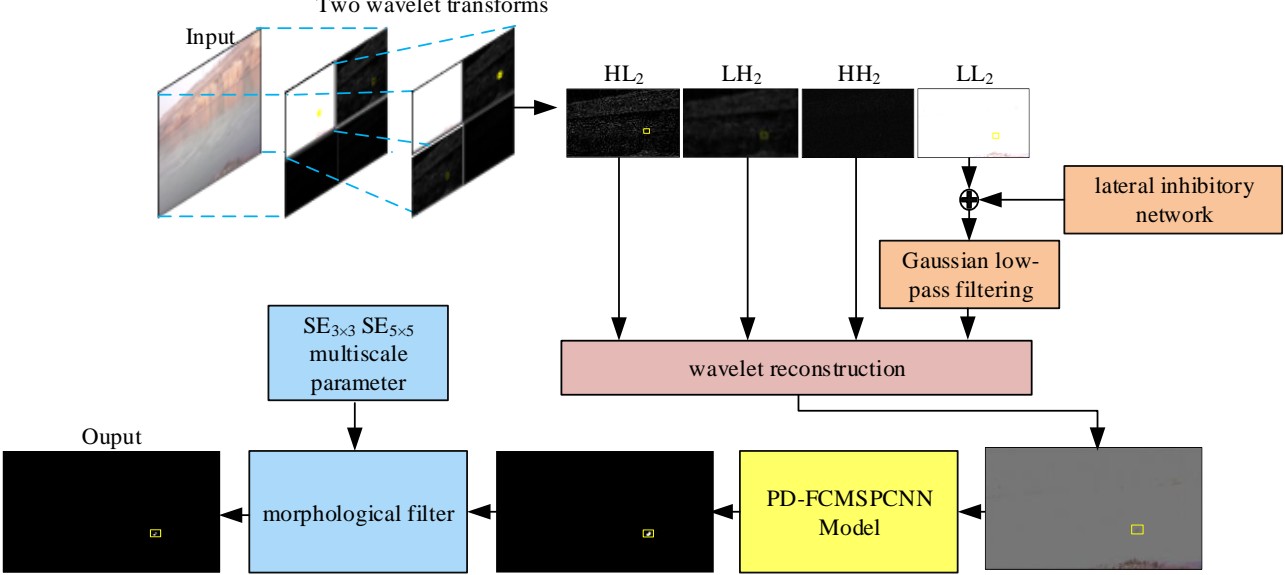

**Figure 3.** Flowchart of proposed algorithm. The yellow box is the marker box for floating garbage.

Step 1: The input image is pre-processed by denoising to obtain high- and low-frequency components after DWT (where $HL_2$, $LH_2$, and $HH_2$ are the high-frequency components in the horizontal, vertical, and diagonal directions after two wavelet transforms, respectively; and $LL_2$ is the low-frequency component after two wavelet transforms). The low-frequency component image is filtered by a low-pass Gaussian filter combined with an LI network, which enhances the edge features of the low-frequency component during denoising. Subsequently, the image is recovered using wavelet inverse transform.

Step 2: The denoised image is input into the PD-FC-MSPCNN segmentation model. The attenuation factor $\alpha$, auxiliary parameter $P$, and amplitude parameter $V$ are calculated and dynamically fine-tuned to generate a synaptic weight matrix $W_{ijkl}$ with a normal distribution for image segmentation.

Step 3: Finally, the segmentation results are processed using multiscale morphological filtering (MMF), which computes the structural elements (SEs) at different scales, connects the segmentation breakpoints, smooths the results, and obtains the target pixel coordinates (upper left, lower right, and center coordinates) to complete the segmentation localization process.

### 2.3. Preprocessing: LI-DWT Denoising

With K input source images, $I_k, k \in [1, K]$ performs DWT decomposition, as shown in Equations (1) and (2):

$$I_k^L(i,j) = \frac{1}{\sqrt{XY}} \sum_{i=0}^{X-1} \sum_{j=0}^{Y-1} I_k(i,j)\varphi(x,y), \qquad (1)$$

$$I_k^G(i,j) = \frac{1}{\sqrt{XY}} \sum_{i=0}^{X-1} \sum_{j=0}^{Y-1} I_k(i,j) \Psi^G(x,y), G \in \{H, D, V\}. \tag{2}$$

where $I_k^L$, $I_k^G$ represents the low-frequency component and high-frequency component, while H, D, and V represent the horizontal, vertical, and diagonal directions. X and Y represent the length and width of image pixels, $(i,j)$ represents the coordinate values of the transformed pixels, $(x,y)$ represents the coordinate values of the input source image pixels, $\varphi(x,y)$ is a scaling function, $\Psi^G(x,y)$ is a wavelet function, and $\varepsilon$ denotes the scaling magnitude.

$$\varphi(x,y) = 2^{\frac{\varepsilon}{2}} \varphi\left(2^{\frac{\varepsilon}{2}}x - i, 2^{\frac{\varepsilon}{2}}y - j\right), \tag{3}$$

$$\Psi^G(x,y) = 2^{\frac{\varepsilon}{2}} \Psi\left(2^{\frac{\varepsilon}{2}}x - i, 2^{\frac{\varepsilon}{2}}y - j\right), G \in \{H, D, V\}. \tag{4}$$

Thus, low- and high-frequency component images can be generated. The low-frequency image contains rich information and is processed by the LI network using Gaussian filtering. Subsequently, a wavelet inversion provides the denoised depth map. The Gaussian filtering proceeds as follows:

$$Q(i,j) = \frac{1}{2\pi\sigma^2} e^{-\frac{i^2+j^2}{2\sigma^2}}, \tag{5}$$

where $\sigma^2$ is the variance, and $m, n$ are the input pixel point coordinates, where $m, n \in [M, N]$. The Gaussian kernel size is $(2t+1) \times (2t+1)$, and in this paper t = 2. The Gaussian kernel function is calculated as

$$Q(i,j) = \frac{1}{2\pi\sigma^2} e^{-\frac{(i-t-1)^2+(j-t-1)^2}{2\sigma^2}}, \tag{6}$$

The LI network is modeled as

$$r_{i,j} = e_{i,j} - \sum_{m=-t}^{m=t} \sum_{n=-t}^{n=t} C_{mm} e(i+m, j+n), \tag{7}$$

where $e_{i,j}$ is the input of a pixel at a point, $r_{i,j}$ is the output of the point, t is the size of the inhibition field, and $C_{mm}$ is the matrix of the LI coefficients, which uses the Euclidean distance $d_{ist} = \sqrt{(i-x)^2 + (j-y)^2}$ between two receptors $a(i,j)b(x,y)$ as the formula for the LI coefficients, such that they have the characteristics of a normal distribution. After calculation and normalization, the parameters are set as follows:

$$C_{mm} = \begin{bmatrix} 0.4000 & 0.3162 & 0.2828 & 0.3162 & 0.0400 \\ 0.3162 & 0.2000 & 0.1414 & 0.2000 & 0.3162 \\ 0.2828 & 0.1414 & 0 & 0.1414 & 0.2828 \\ 0.3162 & 0.2000 & 0.1414 & 0.2000 & 0.3162 \\ 0.4000 & 0.3162 & 0.2828 & 0.3162 & 0.0400 \end{bmatrix}. \tag{8}$$

To enable Gaussian filtering to better handle low-frequency-component images and achieve target edge enhancement, LI is incorporated into the Gaussian kernel as follows:

$$Q_{LI} = P(i,j)^T C_{mm}, \tag{9}$$

Any image from the dataset is tested, and after LI-DWT denoising, the low- and high-frequency components after two wavelet transforms and the wavelet reconstructed image are obtained, as shown in Figure 4.

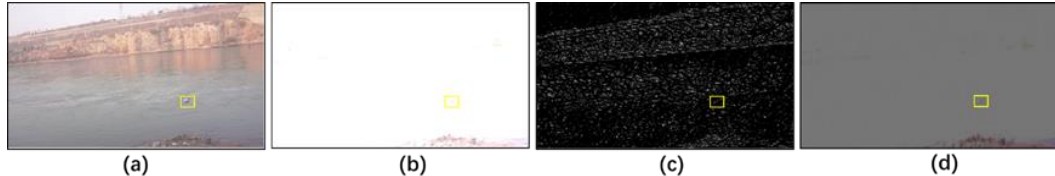

(a)  (b)  (c)  (d)

**Figure 4.** Denoising results of lateral inhibition discrete wavelet transform: (**a**) original figure, (**b**) low-frequency components after two wavelet transforms, (**c**) high-frequency components after two wavelet transforms, and (**d**) wavelet reconstruction after two wavelet transforms. The yellow box is the marker box for floating garbage.

### 2.4. Segmentation: PD-FC-MSPCNN

The PCNN is based on synchronous pulse issuance on the cerebral cortex of cats and monkeys. Compared to deep learning, the PCNN can extract important information from complex backgrounds without learning or training and has the characteristics of synchronous pulse issuance and global coupling. Moreover, its signal form and processing mechanism are more consistent with the physiological basis of the human visual nervous system.

The original PCNN model has complicated parameter settings that require manual settings. The SPCNN model proposed by Chen et al. [32] has been widely used for image segmentation. Its kinetic equations are:

$$F_{i,j}[n] = S_{i,j}, \tag{10}$$

$$L_{ij}[n] = V_L \sum_{kl} W_{ijkl} Y_{kl}[n-1], \tag{11}$$

$$U_{ij}[n] = e^{-\alpha_f} U_{ij}[n-1] + S_{ij}(1 + \beta V_L \sum_{kl} W_{ijkl} Y_{kl}[n-1]), \tag{12}$$

$$Y_{ij}[n] = \begin{cases} 1, if\ U_{ij}[n] > E_{ij}[n-1] \\ 0, else \end{cases}, \tag{13}$$

$$E_{ij}[n] = e^{-\alpha_E} E_{ij}[n-1] + V_E Y_{ij}[n], \tag{14}$$

where $F_{i,j}[n]$, $L_{ij}[n]$, $U_{ij}[n]$, and $E_{ij}[n]$ denote the feedback input, link input, internal activity, and dynamic threshold of the neuron after iteration, respectively. Furthermore, $S_{i,j}$ denotes the input excitations, $V_L$ is the magnitude coefficient of the link input, $V_E$ is the magnitude coefficient of the variable threshold function, $\alpha_f$ and $\alpha_e$ are the decay constants, $\beta$ is the link strength, and $W_{ijkl}$ is the synaptic connectivity coefficient. The five important parameters of the SPCNN model, $W_{ijkl}$, $\alpha_f$, $\beta$, $V_E$, and $\alpha_e$, are set as follows:

$$W_{ijkl} = \begin{bmatrix} 0.5 & 1 & 0.5 \\ 1 & 0 & 1 \\ 0.5 & 1 & 0.5 \end{bmatrix}, \tag{15}$$

$$\alpha_f = log\left(\frac{1}{\sigma(S)}\right), \tag{16}$$

$$\beta = \frac{(S_{max}/\acute{S}) - 1}{6V_L}, \tag{17}$$

$$V_E = e^{-\alpha_f} + 1 + 6\beta V_L, \tag{18}$$

$$V_L = 1, \tag{19}$$

$$\alpha_e = ln\left(\frac{V_E}{\frac{1-e^{-3\alpha_f}}{1-e^{-\alpha_f}} + 6\beta V_L e^{-\alpha_f}}\right). \tag{20}$$

Lian et al. [33] improved the SPCNN and proposed a fire-control MSPCNN model (FC-MSPCNN), which provides a parameter setting method to control the fire-control neurons within an effective pulse period. Consequently, good performance was achieved in color image quantization and gallbladder image localization. In this study, we further simplify the model parameters and computational process based on the FC-MSPCNN model and propose a parameter-designed FC-MSPCNN (PD-FC-MSPCNN), wherein the parameter can be set adaptively, as depicted in Figure 5. First, the two decay constants, $\alpha_f$ and $\alpha_e$, are unified and defined as the parameter $\alpha$. Second, to simplify the operation, the parameter link strength $\beta$ in the internal activity term of $U_{ij}$ is removed. Third, to reduce the complexity that is associated with parameter setting, an auxiliary parameter $P$ and amplitude parameter $V$ are introduced into the dynamic threshold $E_{ij}$ to achieve dynamic fine-tuning of the threshold and to calculate the synaptic weight matrix $W_{ijkl}$ with normal distribution characteristics.

$$U_{ij}[n] = e^{-2\alpha}U_{ij}[n-1] + e^{-\alpha}S_{ij} + \sum_{kl} W_{ijkl}Y_{kl}[n-1]S_{ij}, \tag{21}$$

$$Y_{ij}[n] = \begin{cases} 1, if\ U_{ij}[n] > E_{ij}[n-1] \\ 0, else \end{cases}, \tag{22}$$

$$E_{ij}[n] = e^{-\alpha}E_{ij}[n-1] + P + VY_{ij}[n]. \tag{23}$$

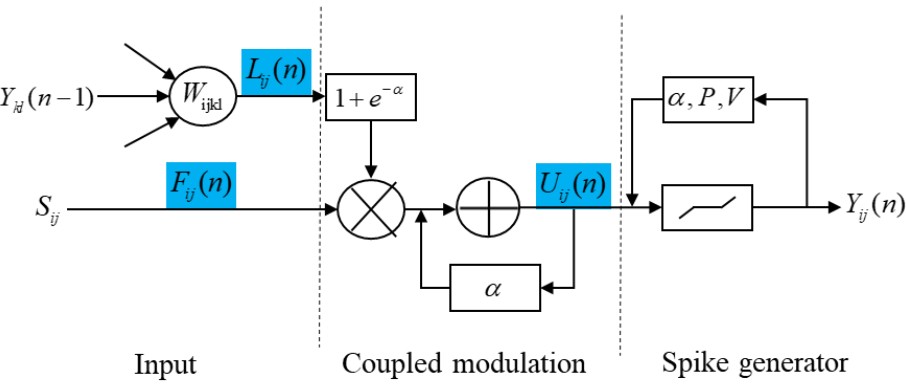

**Figure 5.** PD-FC-MSPCNN structure diagram.

An improved formula for calculating the attenuation factor $\alpha$ is presented below, where $O'$ is the normalized Otsu algorithm segmentation threshold.

$$\alpha = \ln\frac{1}{O'}. \tag{24}$$

The synaptic weight matrix $W_{ijkl}$ is calculated with reference to the calculation provided in the existing literature [33]. In setting this parameter, it highlights that the weight matrix has a normal distribution with a standard deviation of one.

$$W_{ijkl} = \begin{bmatrix} 0.00296 & 0.01331 & 0.02194 & 0.01331 & 0.00296 \\ 0.01331 & 0.05963 & 0.09832 & 0.05963 & 0.01331 \\ 0.02194 & 0.09832 & 0.16210 & 0.09832 & 0.02194 \\ 0.01331 & 0.05963 & 0.09832 & 0.05963 & 0.01331 \\ 0.00296 & 0.01331 & 0.02194 & 0.01331 & 0.00296 \end{bmatrix}. \tag{25}$$

The expressions for the amplitude parameter $V$ and auxiliary parameter $P$ of the dynamic threshold are as follows:

$$V = e^{-\alpha} + e^{-2\alpha} \tag{26}$$

$$P = e^{-3\alpha} + e^{-4\alpha}. \tag{27}$$

The role of the auxiliary parameter $P$ is to fine-tune the threshold dynamically, thereby widening the adjustment range of the PD-FC-MSPCNN model.

The LI-DWT image is fed into the improved PD-FC-MSPCNN model and the segmentation results are obtained, as shown in Figure 6.

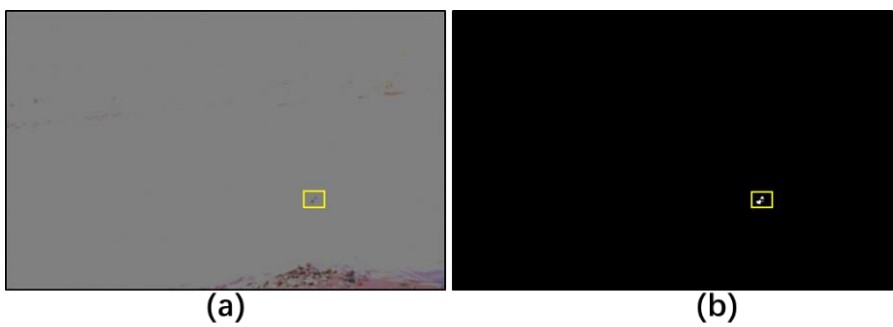

**Figure 6.** PD-FCMSPCNN: (**a**) preprocessing results and (**b**) segmentation results. The yellow box is the marker box for floating garbage.

### 2.5. Postprocessing: MMF

Notably, noise with different scales exists in the segmentation results; the PD-FC-MSPCNN uses morphological filtering for optimization, connection of the breakpoints, and smoothing. The grayscale expansion of the morphological filtering for the segmented image $(i, j)$ is expressed as:

$$(f \oplus SE)(i, j) = max\{f(i - i', j - j') + SE(i', j') | (i', j') \in D_{SE}\}, \tag{28}$$

where $D_{SE}$ is the definition domain of the SE (i.e., a square matrix herein). Subsequently, the grayscale closing morphological filtering operation is defined as

$$f \cdot SE = (f \oplus SE) \ominus SE. \tag{29}$$

This closing operation is performed based on the expansion operation to eliminate darker details that are smaller than the SEs. In general, erosion and the closing operation are mixed and matched to achieve operation filtering effects. However, in certain cases, multiscale noise interference cannot be effectively removed if a single SE is used for filtering. Therefore, to achieve better denoising, we set the SE scales to $3 \times 3$ and $5 \times 5$, and normalize them according to the LI-DWT coefficients and the eight domain correlation properties of the image.

$$SE_{3\times3} = \begin{bmatrix} 1 & 0 & 1 \\ 0 & 1 & 0 \\ 1 & 0 & 1 \end{bmatrix}, \tag{30}$$

$$SE_{5\times5} = \begin{bmatrix} 1 & 0 & 0 & 0 & 1 \\ 1 & 1 & 0 & 1 & 1 \\ 1 & 0 & 1 & 0 & 1 \\ 1 & 1 & 0 & 1 & 1 \\ 1 & 0 & 0 & 0 & 1 \end{bmatrix}. \tag{31}$$

The improved MMF is expressed as follows:

$$f(i, j) = (f \oplus SE_{3\times3})\Theta SE_{5\times5}. \tag{32}$$

The resulting segmentation map is used as the input and MMF is performed. The results are presented in Figure 7.

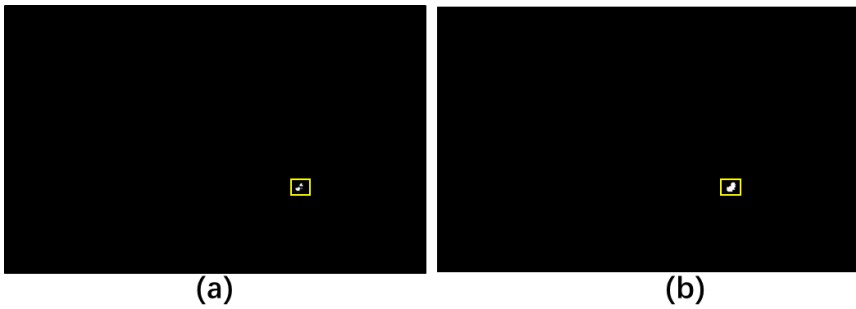

**(a)**　　　　　　　　　　　　　　　　　　**(b)**

**Figure 7.** Multiscale morphological filtering: (**a**) segmentation results and (**b**) morphological filtering. The yellow box is the marker box for floating garbage.

## 3. Results

The experiments are conducted using the deep-learning framework PyTorch, Python 3.8, CUDA 9.0, a GPU of NVIDIA GeForce RTX 2080 Ti, 11 GB of graphics memory, 62 GB of RAM, and Windows 10 as the operating system.

### 3.1. Evaluation Indicators

Eight evaluation metrics, namely the perceptual hash similarity (phash), error rate (VOE), target area variance (RVD), mean absolute error (MAE), Hoffman distance (HF), sensitivity (SEN), time complexity (T), and coordinate error (CD), are selected for this experiment to observe the segmentation extraction results more intuitively. A new overall evaluation score known as the overall comprehensive evaluation (OCE) is set to represent the overall comprehensive evaluation of the algorithm using the eight metrics. The OCE provided the final score based on a particular weight distribution of the eight evaluation metrics.

- Perceived hash similarity (phash):

$$phash(I_{k1}, I_{k2}) = \sum_{i=1}^{z} \frac{\sqrt{\left(i_{1,i} - \rho i_{2,i}\right)^2 + \left(j_{1,i} - \rho j_{2,i}\right)^2}}{Z}, \tag{33}$$

where $X_1$ denotes the segmented image output by the algorithm and $X_2$ denotes the manually segmented image; $\rho = \phi_2(i,j)/\phi_1(i,j)$, $i, j$ denotes the corresponding pixel point. The perceived hash similarity quantifies the overall similarity between the segmented image output by the algorithm and the manually segmented image by calculating the Euclidean distance between the two. A phash value that is closer to one indicates greater similarity between the two inputs.

- Volumetric overlap error (VOE):

$$VOE = \left| \frac{2 \times (I_{k1} - I_{k2})}{I_{k1} + I_{k2}} \right|. \tag{34}$$

The VOE measures the accuracy of the target edge pixel segmentation; a lower error rate indicates that the algorithm pixel segmentation is more accurate and the algorithm is more reliable.

- Relative volume difference (RVD):

$$RVD = \left| \frac{I_{k1}}{I_{k2}} - 1 \right|. \tag{35}$$

The RVD measures the area difference of the segmented image output by the algorithm compared with the manually segmented image; a value that is closer to zero indicates more accurate algorithm segmentation and a better segmentation effect.

- MAE:

$$MAE = \frac{1}{XY} \sum_{i=1}^{X} \sum_{j=1}^{Y} \left( j_{I_{k1}} - j_{I_{k2}} \right) \left( i_{I_{k1}} - i_{I_{k2}} \right). \tag{36}$$

The MAE measures the deviation of the segmented image output by the algorithm from the manually segmented image. A smaller MAE value indicates a smaller overall error in the segmentation result output by the algorithm compared with the manually segmented image.

- Hausdorff distance (HF):

$$HF = \max(h(I_{k1}, I_{k2}), h(I_{k2}, I_{k1})). \tag{37}$$

The HF indicates the maximum distance between two sets of segmented images output by the algorithm and manually segmented images; a smaller HF value indicates that the segmentation results output by the algorithm are closer to the manually segmented images.

- Sensitivity (SEN):

$$SEN = \frac{I_{k1} \cap I_{k2}}{I_{k2}}. \tag{38}$$

The SEN indicates the ratio of the algorithm segmentation relative to the manual segmentation results and that the region of the target is correctly judged as the target. SEN values close to one indicate a good overall applicability of the algorithm and high segmentation accuracy on different data types.

- Time complexity (T):

T quantifies the efficiency of the algorithm in terms of time; time complexity in deep learning determines the predictive inference speed of the network.

- Coordinate distance error (CD):

$$CD = \sum_{n=1}^{3} \sqrt{ \left( i_{I_{k1,n}} - i_{I_{k1,n0}} \right)^2 + \left( j_{I_{k1,n}} - j_{I_{k1,n0}} \right)^2 }, \tag{39}$$

where $i_n$, $j_n$ indicate the pixel coordinates of the target output by the algorithm; $n = 1$ is the upper-left pixel coordinate of the target, $n = 2$ is the lower right pixel coordinate of the target, $n = 3$ is the center pixel coordinate of the target, and $i_{n0}$, $j_{n0}$ indicate the pixel coordinates of the target point output by the manual marker. The CD measures the localization accuracy of the algorithm, where smaller values indicate higher localization accuracies.

- Overall comprehensive evaluation (OCE):

We demonstrate that the overall segmentation achieved using the proposed algorithm was superior to that achieved by conventional strategies. Accordingly, the above eight evaluation indexes were normalized separately and combined into the OCE.

$$OCE = \frac{1}{3}(1 - T) + \frac{1}{3}\left( \frac{phash + SEN}{2} \right) + \frac{1}{3}\left[ \frac{(1 - VOE) + (1 - RVD) + (1 - MAE) + (1 - HF) + (1 - CD)}{5} \right]. \tag{40}$$

The OCE comprises three sets of equally weighted metrics that are divided into the overall error, overall similarity, and time complexity scores. The overall error score consists of 1-VOE, 1-RVD, 1-MAE, 1-HF, and 1-CD with the weight set to $\frac{1}{5}$ to verify the overall image segmentation error. The overall similarity score comprises phash and the SEN with the weight set to $\frac{1}{2}$ to evaluate the overall similarity of the segmented images output by the algorithm to the manually segmented images. The time complexity score comprises $1 - T$ and measures the inference speed of the algorithm. Notably, phash, VOE, RVD, MAE, HF, SEN, T, and CD in Equation (40) are linearly normalized values.

### 3.2. Ablation Experiments

Ablation experiments are performed using dataset 1 with the following configurations: (1) LI-DWT + PD-FCMSPCNN, (2) PD-FCMSPCNN + MMF, and (3) LI-DWT + PD-FCMSPCNN + MMF, as illustrated in Figure 8, to quantify the modular contributions of the model components. The comparison results are shown in Table 1.

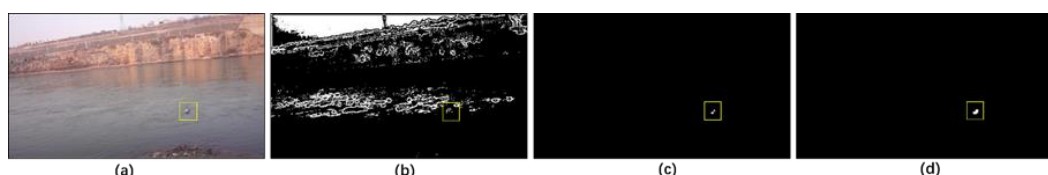

**Figure 8.** Comparison graph of ablation experiments: (**a**) original graph, (**b**) experimental results of group (1), (**c**) experimental results of group (2), and (**d**) experimental results of group (3). The yellow box is the marker box for floating garbage.

**Table 1.** Comparison of index ablation for different strategies.

| Method | Phash | VOE | RVD | MAE | HD | SEN |
| --- | --- | --- | --- | --- | --- | --- |
| (1) | 89.97% | 0.1389 | 0.1679 | 0.1492 | 15.03 | 0.8382 |
| (2) | 57.81% | 1.8923 | 35.1342 | 0.4220 | 144 | 0.0271 |
| (3) Ours | 95.32% | 0.0322 | 0.0323 | 0.0017 | 7.37 | 0.9432 |

A comparison of configurations 3 and 2 reveals that the preprocessing method of LI-DWT significantly improves the segmentation results; in particular, the phash value increases from 57.81% to 95.32%, VOE decreases from 1.8923 to 0.0322, RVD decreases from 35.1342 to 0.0323, MAE decreases from 0.422 to 0.0017, HD decreases from 144 to 7.37, and SEN increases from 0.0271 to 0.9432. These results indicate that the preprocessing method can effectively filter out interference from illumination, water waves, and complex backgrounds, thereby making the segmentation results more consistent with the visual characteristics of human eyes and improving the segmentation accuracy. A comparison of configurations 3 and 1 demonstrates that the phash value increases from 89.97% to 95.32%, VOE decreases from 0.1389 to 0.0322, RVD decreases from 0.1679 to 0.0323, MAE decreases from 0.1492 to 0.0017, HD decreases from 13.03 to 7.37, and SEN increases from 0.8382 to 0.9432. These results indicate that the postprocessing method of MMF also improves the segmentation effect. Thus, the rationality and reliability of the proposed method are verified through the ablation experiments.

### 3.3. Comparative Analysis of Split Extraction Algorithms

The proposed method is compared with the Transformer-based U-Net (TransUNet) framework, a lightweight DeepLabv3+ model, the high-resolution medical image segmentation network (HRNet), and the lightweight pyramid scene-parsing network (PSPNet). Figure 9a–g presents the plots of the segmentation results for the original image using PSPNet, DeepLabv3+, HRNet, TransUNet, the proposed method, and manual labeling, respectively; these segmentation results correspond to the application of different algorithms on the three datasets.

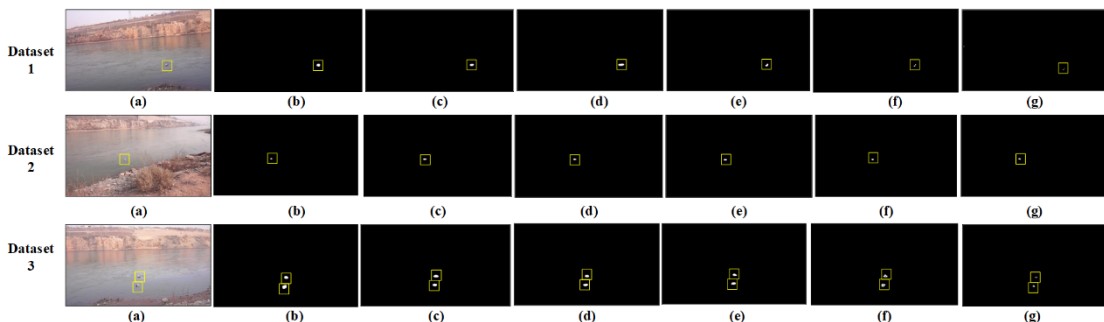

**Figure 9.** Algorithms used for comparison on three types of datasets: (**a**) original image, (**b**) PSPNet, (**c**) DeepLabv3+, (**d**) HRNet, (**e**) TransUNet, (**f**) ours, and (**g**) manual marker. The yellow box is the marker box for floating garbage.

The evaluation results of the different algorithms used for comparison on the three data sets are listed in Table 2. In dataset 1, the evaluation indicators are as follows: the phash value of the proposed algorithm is 95.32%, which is better than the phase values of the comparison algorithms, indicating that the segmentation results of the proposed method are most similar to the manual segmentation results. The VOE value of the proposed algorithm is 0.0322, which is lower than the VOE value of the comparison algorithms, indicating that the segmentation error rate of the proposed method is lower than the comparison algorithms and the proposed method has the best target edge pixel segmentation ability. The RVD value of the proposed algorithm is 0.0323, which is lower than the RVD value of the comparison algorithms, indicating that the target segmentation area of the proposed method is the closest to the manual target segmentation area and the proposed method achieves better target edge differentiation. The MAE value of the proposed algorithm is 0.0017, which is lower than the MAE value of the comparison algorithm, indicating that the segmentation result of the proposed method has the smallest average absolute error compared with the manual segmentation result and the proposed method has the best target edge differentiation quality. The SEN value of the proposed algorithm is 0.9432, which is better than the SEN value of the comparison algorithms, indicating that the target segmentation result of the proposed method has the smallest intersection ratio with the manual target segmentation result; furthermore, the boundary of the proposed algorithm is sensitive and the edge segmentation effect is optimal. The CD value of the proposed algorithm is 4.38, which is lower than the CD value of the comparison algorithms, indicating that the proposed method has the smallest localization error. Moreover, the values of HD for HRNet (6.71) and the T value for DeepLabv3+ (84.6) are slightly better than those of the proposed method. However, the remaining indexes for the proposed method are better than those of the comparison algorithms.

For dataset 2, the VOE, RVD, MAE, SEN, and CD values of the proposed method are 0.0466, 0.0476, 0.0015, 0.9224, and 4.47, respectively, which are all outperform the values of the comparison algorithm. Although the values of phash (92.83%), HD (7.57), and T (163.7) of the proposed method are slightly inferior to those of the comparison algorithms, in general, the proposed algorithm achieves more accurate segmentation of the target area edges in dataset 2, segmented the target areas closer to the manually segmented area, and achieved the lowest error rates and high segmentation efficiency while maintaining low complexity; thus, it is more conducive to high-precision target positioning.

The data of dataset 3 comprised multiple objectives, which causes a certain degree of error accumulation; thus, the proposed algorithm exhibits lower values of phash, VOE, RVD, MAE, HD, and SEN compared to its values for datasets 1 and 2. The values of phash, VOE, RVD, SE, and CD of the proposed algorithm are 92.31%, 0.0615, 0.0648, 0.9168, and 4.56, respectively, which all outperform the values of the comparison algorithms. However, the MAE is 0.0028, HD is 8.24 and T is 164.0, which is between the MAE, HD, and T values of the comparison algorithms. Overall, the proposed algorithm has better segmentation

results that are closer to the manual segmentation results with the minimum computational time and localization error.

**Table 2.** Comparison of evaluation results of different algorithms.

| Dataset | Method | Phash | VOE | RVD | MAE | HD | SEN | T (ms) | CD |
|---------|--------|-------|-----|-----|-----|----|----|--------|-----|
| Dataset 1 | PSPNet | 91.34% | 0.0409 | 0.0426 | 0.0018 | 7.94 | 0.9283 | 107.2 | 8.63 |
| | DeepLabv3+ | 92.21% | 0.0428 | 0.0438 | 0.0022 | 7.73 | 0.9290 | 84.6 | 6.88 |
| | HRNet | 91.40% | 0.0557 | 0.0585 | 0.0023 | 6.71 | 0.8954 | 166.7 | 8.50 |
| | TransUNet | 94.91% | 0.0406 | 0.0428 | 0.0018 | 7.19 | 0.9343 | 211.1 | 5.75 |
| | Ours | 95.32% | 0.0322 | 0.0323 | 0.0017 | 7.37 | 0.9432 | 163.1 | 4.38 |
| Dataset 2 | PSPNet | 91.85% | 0.0495 | 0.0517 | 0.0017 | 7.90 | 0.9063 | 108.6 | 8.71 |
| | DeepLabv3+ | 91.85% | 0.0542 | 0.0562 | 0.0018 | 7.26 | 0.9103 | 88.5 | 6.92 |
| | HRNet | 92.55% | 0.0468 | 0.0484 | 0.0016 | 7.85 | 0.9093 | 164.2 | 8.54 |
| | TransUNet | 94.16% | 0.0489 | 0.0491 | 0.0016 | 6.83 | 0.9216 | 212.3 | 5.83 |
| | Ours | 92.83% | 0.0466 | 0.0476 | 0.0015 | 7.57 | 0.9224 | 163.7 | 4.47 |
| Dataset 3 | PSPNet | 91.71% | 0.1069 | 0.1131 | 0.0029 | 8.35 | 0.8627 | 107.9 | 8.67 |
| | DeepLabv3+ | 91.71% | 0.0867 | 0.0907 | 0.0028 | 8.16 | 0.8703 | 89.3 | 6.90 |
| | HRNet | 91.82% | 0.0803 | 0.0812 | 0.0028 | 8.12 | 0.8814 | 166.2 | 8.63 |
| | TransUNet | 92.27% | 0.0711 | 0.0726 | 0.0025 | 8.06 | 0.8943 | 211.4 | 5.60 |
| | Ours | 92.31% | 0.0615 | 0.0648 | 0.0028 | 8.24 | 0.9168 | 164.0 | 4.56 |

## 4. Discussion

In this research, we validate the effectiveness and reliability of the proposed algorithm in the field of small target floating garbage localization through ablation experiments and comparative experiments with other mainstream algorithms.

In the ablation experiments, we observe that the LI-DWT preprocessing method and MMF post-processing method play a significant role in filtering noise, improving image quality, and enhancing segmentation effect, significantly improving the accuracy of small target water-floating garbage localization. Especially, LI-DWT improves the phase and SEN values, reduces VOE, RVD, MAE and HF values. These results indicate that LI-DWT can effectively filter out interference from light, water waves, and complex backgrounds, making the segmentation results more in line with human visual characteristics and improving segmentation accuracy. The MMF post-processing method can smooth the segmentation results of PD-FC-MSPCNN and improve the accuracy of small target floating garbage localization.

In order to better unify and verify the comprehensive performance of the proposed algorithm, a new OCE index is designed for comparative experiments. The values of OCE is calculated according to Equation (40) are depicted in Figure 10. The OCE values of the proposed method are 0.7574, 0.6555, and 0.7074 on the three datasets, which are better than those of the comparison algorithms. Owing to the normalized index, the variability between algorithms is large. The superiority of the proposed algorithm is measured by the OCE according to three aspects: the overall error score, overall similarity score, and time complexity score. From our experimental results, it is apparent that the proposed algorithm exhibits negligible difference in the segmentation localization performance between single and multiple target scenarios. This claim is supported by the minimal disparity in OCE values observed between datasets 1 and 3, suggesting the proposed algorithm's proficient adaptation to the positioning requirements of diverse target scenes. Interestingly, the positioning precision for a single target appears superior to that of multiple targets. This phenomenon can be attributed to possible interference among targets in multiple target scenarios, warranting further investigation in our future work. When applied to dataset 2, the OCE values of our algorithm revealed significant discrepancies as opposed to the results obtained from datasets 1 and 3. This compellingly underlines the profound impact of intricate background interference on the positioning precision of our proposed algorithm. Overall, the proposed method exhibits the advantages of accurate edge segmentation, a small error, a low computational complexity, and high localization accuracy on three

datasets. It shows good evaluation results, thereby effectively overcoming the interference of complex backgrounds, illumination, and water waves. Furthermore, it could more effectively solve the problem of small target floating garbage on water surfaces with low localization accuracy. In subsequent research, our aim will be to further refine the proposed algorithm, with the aim of mitigating the disturbance caused by intricate backgrounds and enhance the comprehensive interference-resistant capabilities of the algorithm. This will involve exploring sophisticated strategies for both pre-processing and post-processing to augment the algorithm's adaptability across varied complex scenarios, thereby bolstering its robustness.

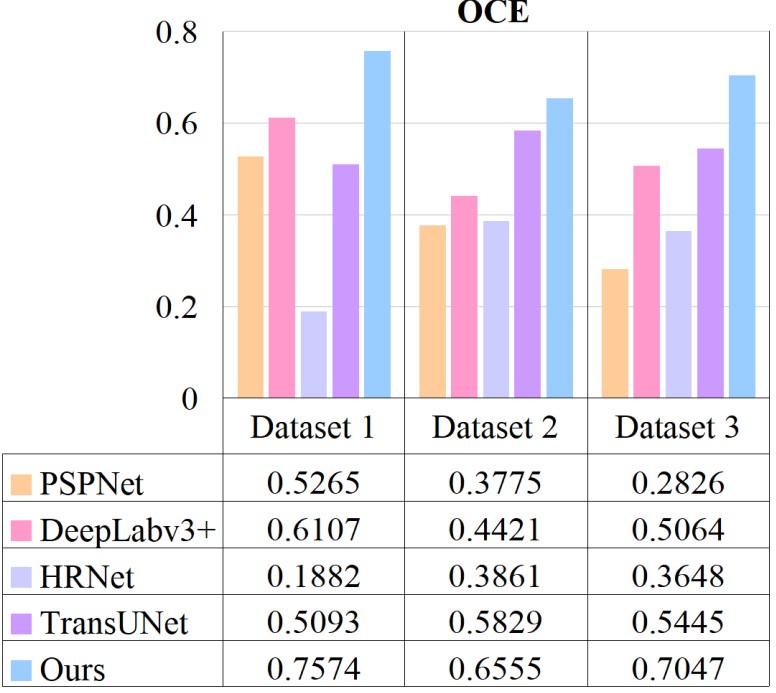

| | Dataset 1 | Dataset 2 | Dataset 3 |
|---|---|---|---|
| ■ PSPNet | 0.5265 | 0.3775 | 0.2826 |
| ■ DeepLabv3+ | 0.6107 | 0.4421 | 0.5064 |
| ■ HRNet | 0.1882 | 0.3861 | 0.3648 |
| ■ TransUNet | 0.5093 | 0.5829 | 0.5445 |
| ■ Ours | 0.7574 | 0.6555 | 0.7047 |

**Figure 10.** OCE indicator evaluation results.

## 5. Conclusions

A small-target image segmentation and localization method based on the LI-DWT and PD-FC-MSPCNN has been proposed for water-floating garbage. This method improves the localization accuracy of the intelligent tracking and position prediction of small-sized floating garbage. By integrating LI with the DWT, the method reduces the interference of illumination, water waves, and complex backgrounds on the image segmentation. The proposed PD-FC-MSPCNN segmentation model efficiently achieves high-precision small-target segmentation. Moreover, the improved MMF connects the segmentation breakpoints and smoothens the segmentation results.

Nine evaluation metrics and four algorithms for comparison are selected for the experimental analysis on three different datasets collected from the Lanzhou section of the Yellow River. The OCE metrics of the proposed method on the three datasets are 0.7574, 0.6555, and 0.7074. The PD-FC-MSPCNN model achieve optimal segmentation results as well as high localization accuracy and good computational performance. The proposed method uses floating garbage on water surfaces as the study object and does not consider other types of floating objects on water surfaces, such as ships, floating duckweed and oil slicks. In the future, we will improve algorithms to improve positioning accuracy in complex backgrounds, and expand the research category of floating debris on the water surface, further improving the applicability and generalization ability of the algorithm.

**Author Contributions:** Conceptualization, P.A. and L.M.; Methodology, P.A., L.M. and B.W.; Software, B.W.; Validation, P.A., L.M. and B.W.; Formal Analysis, L.M.; Investigation, L.M. and B.W.; Resources, L.M.; Data Curation, B.W.; Writing—Original Draft Preparation, L.M.; Writing—Review and Editing, L.M.; Visualization, L.M.; Supervision, P.A.; Project Administration, L.M.; Funding Acquisition, L.M. All authors have read and agreed to the published version of the manuscript.

**Funding:** This research is supported by 'The project of Intelligent identification and early warning technology of garbage in river and lake shoreline of Department of Water Resources of Gansu Province' under Grant LZJT20221013.

**Data Availability Statement:** Data used in this study is available upon request to the corresponding author. Code is available at https://github.com/jingcodejing/PD-FCM-SPCNN accessed on 3 June 2023.

**Acknowledgments:** Thanks are due to Jing Lian for valuable discussion. Funding from the Department of Water Resources of Gansu Province is gratefully acknowledged.

**Conflicts of Interest:** The authors declared no potential conflict of interest with respect to the research, authorship, and/or publication of this article.

## Abbreviations

| | |
|---|---|
| LI | lateral inhibition network |
| DWT | discrete wavelet transform |
| MMF | multi-scale morphological filtering |
| FC-MSPCNN | a fire-controlled MSPCNN model |
| PD-FC-MSPCNN | parameter-designed-FC-MSPCNN |
| Faster R-CNN | faster Regions with Convolutional Neural Network |
| CA | Class Activation |
| Yolo | You Only Look Once |
| CNN | Convolutional Neural Network |
| GAN | Generative Adversarial Network |
| PCNN | Pulse Coupled Neural Network |
| SPCNN | simplified PCNN |
| SCM | spiking cortical model |
| SCHPCNN | oscillating sine–cosine pulse coupled neural network |
| HVS | Human Vision System |
| SM-ISPCNN | saliency motivated improved simplified pulse coupled neural network |

## Appendix A

**Algorithm A1** A Small Target Location Method for Floating Garbage on Water Surface Based on LI-DWT and PD-FC-MSPCNN
Implementation Steps

| | |
|---|---|
| Input | Color image of floating garbage on water surface, image size x × y, normalized Otsu threshold $S'$, number of iterations t, structure element SE. |
| Pre-processing | For: T = 2<br>Generating high- and low-frequency component images by T times discrete wavelet transform by (1) and (2).<br>By (3) and (4), the Gaussian low-pass filter is $G(u, v)$.<br>  For: f(u, v) = 5 × 5<br>Using (5), the Gaussian kernel f(u, v) is calculated, and the lateral inhibition network is introduced using (6)–(9), the lateral inhibition network is fused, and the wavelet reconstruction outputs the denoised image.<br>  End<br>End |

| | |
|---|---|
| | **Algorithm A1** *Cont.* |
| Segmentation | For i = 1:x<br>  For j = 1:y<br>  Using (21)–(23), the parameter values of PD-FCMSPCNN are calculated, including feed input, link input, internal activity, excitation state, and dynamic threshold.<br>  End<br>End<br>For t = 30<br>Using (24)–(27), the attenuation factor $\alpha$, weight matrix $W_{ijkl}$, magnitude parameter $V$ of the dynamic threshold, and the auxiliary parameter $P$ are calculated.<br>If   $U_{ij}[n] > E_{ij}[n]$<br>$$Y_{ij}[n] = 1$$<br>Else $Y_{ij}[n] = 0$<br>End<br>By (23), the dynamic threshold is calculated $E_{ij}[n]$.<br>End |
| Post-processing | For $SE = 3 \times 3$<br>Set by (30) $SE_{3\times3}$<br>For $SE = 5 \times 5$<br>Set by (31) $SE_{5\times5}$<br>End<br>Using (32), the morphological filtering results are calculated<br>Calculate the pixel coordinates of the segmented target (top left, bottom right and center)<br>End |
| Output | Image and coordinates of floating garbage segmentation results on water surface |

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
