# Peer review of "LI-DWT- and PD-FC-MSPCNN-Based Small-Target Localization Method for Floating Garbage on Water Surfaces"

_water, doi:10.3390/w15122302_

Round 1

Reviewer 1 Report

This is an interesting study. The methods, results, and conclusion have scientific merit. However, I recommend them to share the code on GitHub for reproducibility.

Author Response

Response to Reviewer 1 Comments

Dear reviewers:

Thank you very much for providing valuable review comments for the article we submitted. We attach great importance to your review conclusion and have made modifications based on your suggestions to make the article more complete and rigorous. For the convenience of review, the red font is used to explain the reasons and methods for the modification, while the blue font is used for the specific content of the modification. The modifications are as follows:

Point 1: Comments and Suggestions for Authors:

This is an interesting study. The methods, results, and conclusion have scientific merit. However, I recommend them to share the code on GitHub for reproducibility.

Response 1:

Thank you to the reviewer for taking the time to review our manuscript. In response to your feedback, we have made the code publicly available at https://github.com/jingcodejing/PD-FCM-SPCNN and provided annotations and explanations in the manuscript.

Codes are available at https://github.com/jingcodejing/PD-FCM-SPCNN.

Reviewer 2 Report

Thanks, the authors for submitting their works to the Journal.

Authors present LI-DWT- and PD-FC-MSPCNN-based small-target localization 2 method for floating garbage on water surfaces.

Discussion is to short.

Author Response

Dear reviewers:

Thank you for your suggestions on the completeness of the experimental analysis. For the convenience of review, the red font is used to explain the reasons and methods for the modification, while the blue font is used for the specific content of the modification. The modifications are as follows:

Point 1: Comments and Suggestions for Authors:

Thanks, the authors for submitting their works to the Journal.

Authors present LI-DWT- and PD-FC-MSPCNN-based small-target localization 2 method for floating garbage on water surfaces.

Discussion is to short.

Response 1:

Thank you for reading our manuscript and providing valuable feedback. We have carefully read and analyzed your feedback and made revisions to it. In the discussion section, We demonstrated the effectiveness of the pre-processing method and post-processing method through ablation experimental analysis, and the superiority of the proposed method was demonstrated through comparative experiments. At the same time, the limitations and shortcomings of this study were pointed out, as well as prospects for future research directions and content. Annotations were made in the original text. The revised section is as follows:

In this research, we validate the effectiveness and reliability of the proposed algorithm in the field of small target floating garbage localization through ablation experments and comparative experiments with other mainstream algorithms.

In the ablation experiment, we observe that the LI-DWT preprocessing method and MMF post-processing method play a significant role in filtering noise, improving image quality, and enhancing segmentation effect, significantly improving the accuracy of small target water-floating garbage localization. Especially, LI-DWT improves the phase and SEN values, reduces VOE, RVD, MAE and HF values. These results indicate that LI-DWT can effectively filter out interference from light, water waves, and complex backgrounds, making the segmentation results more in line with human visual characteristics and improving segmentation accuracy. The MMF post-processing method can smooth the segmentation results of PD-FC-MSPCNN and improve the accuracy of small target floating garbage localization.

In order to better unify and verify the comprehensive performance of the proposed algorithm, a new OCE index is designed for comparative experiments. The values of OCE is calculated according to Equation (40) are depicted in Figure 10. The OCE values of the proposed method are 0.7574, 0.6555, and 0.7074 on the three datasets, which are better than those of the comparison algorithms. Owing to the normalized index, the variability between algorithms is large. The superiority of the proposed algorithm is measured by the OCE according to three aspects: the overall error score, overall similarity score, and time complexity score. From our experimental results, it is apparent that the proposed algorithm exhibits negligible difference in the segmentation localization performance between single and multiple target scenarios. This claim is supported by the minimal disparity in OCE values observed between datasets 1 and 3, suggesting the proposed algorithm's proficient adaptation to the positioning requirements of diverse target scenes. Interestingly, the positioning precision for a single target appears superior to that of multiple targets. This phenomenon can be attributed to possible interference among targets in multiple target scenarios, warranting further investigation in our future work. When applied to dataset 2, the OCE values of our algorithm revealed significant discrepancies as opposed to the results obtained from datasets 1 and 3. This compellingly underlines the profound impact of intricate background interference on the positioning precision of our proposed algorithm. Overall , the proposed method exhibits the advantages of accurate edge segmentation, a small error, a low computational complexity, and high localization accuracy on three datasets. It shows good evaluation results, thereby effectively overcoming the interference of complex backgrounds, illumination, and water waves. Furthermore, it could more effectively solve the problem of small target floating garbage on water surfaces with low localization accuracy. In subsequent research, our aim will be to further refine the proposed algorithm, with the intention to mitigate the disturbance caused by intricate backgrounds and enhance the comprehensive interference-resistant capabilities of the algorithm. This will involve exploring sophisticated strategies for both pre-processing and post-processing to augment the algorithm's adaptability across varied complex scenarios, thereby bolstering its robustness.

Reviewer 3 Report

In this paper, a small target location method for floating garbage on water surface, is proposed. 

The entire pipeline of the proposed methodology consists of 3 steps:

A pre-processing step, where denoising of the images is performed based on the DWT transformation.

The main step of the algorithm, where a PCNN-type deep network implements the segmentation of the images. In this step, the desired targets are identified.

A post-processing step, where processing of the target objects is performed to complete the segmentation localization process.

The paper is well-written, the methodology includes several innovative elements, and the results are satisfactory, within the context of an extensive experimentation.

Therefore, I recommend accepting the paper with the following clarifications and revisions:

1) In section 2.3, is a 2D DWT transformation applied to the input images? If yes, there is no need to present the 1D equations of the corresponding transformation 

(Equations 1 and 2). You can directly present the relevant equations of the 2D DWT transformation or clarify more explicitly that separate 1D DWT transformations are performed on the rows and columns of the images.

2) In my opinion, equations 5, 7, 9, and 25, which present specific numerical results of matrices used in the paper, can be omitted.

3) In line 224, in the formula for the Euclidean distance, the exponent 2 should be placed outside the parentheses.

4) Line 83: it increase -> it increases

5) Line 286: closure -> closing and line 287: closed -> closing.

Round 2

Reviewer 1 Report

My comments are addressed I recommend to accept this paper.

Reviewer 3 Report

All of my suggested recommendations have been taken into consideration by the authors in the new version of the paper, and therefore, I recommend accepting it.